# The Case of a 44-Year-Old Survivor of Unrepaired Tetralogy of Fallot, Right Aortic Arch and Abdominal Aortopulmonary Collateral Vessels

**DOI:** 10.3390/medicina58081011

**Published:** 2022-07-28

**Authors:** Roxana Ciltea, Alina Ioana Nicula, Mircea Bajdechi, Alexandru Scafa-Udriste, Roxana Rimbas, Gheorghe Iana, Dragos Vinereanu

**Affiliations:** 1Radiology and Medical Imaging Department, University of Medicine and Pharmacy Carol Davila, 050474 Bucharest, Romania; roxanaciltea@yahoo.com (R.C.); george_iana@yahoo.com (G.I.); 2University Emergency Hospital of Bucharest, 050098 Bucharest, Romania; roxanasisu@gmail.com (R.R.); vinereanu@gmail.com (D.V.); 3Doctoral School of Medicine, “Ovidius” University, 900470 Constanta, Romania; mircea.bajdechi@gmail.com; 4Cardiology and Cardiovascular Surgery Department, University of Medicine and Pharmacy Carol Davila, 050474 Bucharest, Romania; alexscafa@yahoo.com

**Keywords:** tetralogy of fallot, major aortopulmonary collateral arteries, cardiac magnetic resonance, computer tomography angiography

## Abstract

The most common congenital cyanotic heart disease is described in the literature as the Tetralogy of Fallot. This abnormality is characterized by the presence of ventricular septal defect (VSD), obstruction of the right ventricular (RV) outflow tract, right ventricular hypertrophy, and overriding aorta. In patients with pulmonary atresia with ventricular septal defect (PA/VSD), major aortopulmonary collateral arteries (MAPCA) are common; however, although some of them do not have PA/VSD, they do have other particular anatomical variants. The case we are presenting in this article is a rare mild symptomatic adult noncorrected TOF, with preserved RV function, right aortic arch, and MAPCAs (“classic” thoracic MAPCAs but also abdominal MAPCAs). The anatomy of a complex congenital defect is well illustrated by cardiac magnetic resonance (CMR) and computer tomography angiography (CTA), and these imaging techniques are mostly used to understand the relative clinical “silence” TOF. Imaging scans thus play a key role in the evaluation of these patients, being very important to know the indications and limitations of each method, but also to learn to combine them with each other depending on the clinical picture of the patient’s presentation. Additionally, the close collaboration between clinicians and imagers is essential for a correct, complete and detailed preoperative evaluation, being subsequently essential for cardiovascular surgeons, the whole team thus deciding the best therapeutic management.

## 1. Introduction

Tetralogy of Fallot (TOF) is one of the most commonly encountered congenital heart malformations, comprising a ventricular septal defect (VSD), right ventricular (RV) outflow tract obstruction, RV hypertrophy, and overriding aorta [1,2,3]. Major aortopulmonary collateral arteries (MAPCAs) are frequently found in association with pulmonary atresia with ventricular septal defect (PA/VSD) [4,5,6,7] However, some patients with MAPCAs do not have PA/VSD but have a variety of other “atypical” anatomic diagnoses [5].

Advances in current surgical techniques in the repair of congenital anomalies facilitate the survival of patients an ensure a good quality of life for them [8]. Imaging examinations have an essential role in diagnosing these patients, thus it becomes very important to know and understand the limits of each imaging method as well as the need to “combine” these diagnostic tools one with each other depending on the clinical spectrum of the patient.

Cardiac magnetic resonance (CMR) is an excellent noninvasive tool because of its accuracy in tissue characterization in various cardiomyopathies such as dilated cardiomyopathy or hypertrophic cardiomyopathies, left ventricular hypertrophy (such as hypertension-induced cardiomyopathy, familial hypertrophic cardiomyopathy, amyloidosis) [9] and also can provide vital details allowing a stepwise approach to correctly and completely define cardiac anomalies of the TOF. Moreover, Coronary CT angiography (CTA) had already proved to be useful in the assessment of the aortopulmonary collateral vessels, which develop to supply under-perfused areas of the pulmonary bed [10] and pose a unique and challenging problem at the time of surgical repair. Both imaging techniques give us complementary information about the anatomy and function of the TOF, helping in planning surgical management [11,12].

## 2. Case Report

A 44-year-old Caucasian man working on a construction site was sent by his employer for an annual cardiac clinical examination. No suspicion was raised at previous routine examinations. From his antecedents, the patient described a minimally symptomatic childhood, dyspnea and cyanosis appearing only when exerting intense effort. The clinical examination showed a slight kyphosis and larger intercostal spaces on the left side (these determine the secondary asymmetry of the rib cage), a lower sternal fusion defect (corresponding to the xiphoid process) and finger clubbing (Figure 1). An anterior chest systolic murmur suggestive for ventricular septal defect was detected. The laboratory tests performed showed significant polycythemia.

All signs of this first assessment suggested a potentially cyanogenic congenital heart disease, the most probable diagnosis being TOF. However, most children with TOF have important central cyanosis and severe health problems derived from hypoxemia. Our patient presented no such symptoms. It is worth noticing that only a few children with TOF do not turn blue at all, especially if the pulmonary stenosis is mild, the ventricular septal defect is small, or both.

In some cases, the cyanosis is quite subtle and may go undetected for some time, such as in our case.

At the time of the initial cardiac evaluation, the patient received beta blocker for tachycardia. When a congenital disease is suspected, the next step is to perform a transthoracic echocardiography (TTE).

### 2.1. Initial Work Up

The ECG showed sinus tachycardia and deviation of the right axis. Trans-thoracic ultrasound, despite the extremely difficult acoustic window due to the asymmetry of the rib cage, showed the presence of a perimembranous ventricular septal defect, the slight hypertrophy of RV, LV and RV being enlarged but with preserved function. Due to the poor acoustic window, the patient was referred to other imaging evaluations to optimally evaluate and characterize the complex cardiovascular anatomy.

### 2.2. Diagnosis and Management

It is known that TOF has a broad anatomical spectrum. Before corrective surgery, the factors that need to be determined are the development of the pulmonary arteries, the levels and degree of obstruction of the right ventricular outflow tract and the presence of collateral arteries [13]. CMR allows excellent three-dimensional anatomical analysis, and unlike transthoracic ultrasound, is not limited by body size or acoustic window.

Thus, at CMR steady-state free precession cine sequences (Figure 2) demonstrated the preservation of left ventricle (LV) function and showed perimembranous ventricular septal defect detected on transthoracic ultrasound, 15 mm wide; the ascending segment of the thoracic aorta was overriding the interventricular septum, and the descending segment has a right paravertebral trajectory, creating a typical right aortic arch look; all segments of the aorta had a normal diameter. RV also had preserved function but was hypertrophied (parietal thickness 8 mm) and had hypertrophied infundibular muscle with right ventricular outflow tract stenosis. Tricuspid regurgitation was mild. Delayed contrast enhanced images (LGE—Figure 3) revealed the presence of a nodular contrast enhancement at the level of the posterior septal insertion of the RV free wall, suggesting increased intracavitary pressures in this case.

However, to complement the CMR, a CTA was requested to assess the collateral vessels for pulmonary supply in more detail (Figure 4). Thus, the presence of a right aortic arch was highlighted, aorta having normal dimensions. An important periesophageal, peribronchial, intercostal collateral circulation (especially on the right side) was visualized, and the right mammary artery was prominent. The main pulmonary artery (PT) had a small caliber (with an ascending aortic/pulmonary trunk ratio of 3.6), associating a small diameter of the origin of both pulmonary arteries. A sacciform ectasia was highlighted on the anterior contour of the proximal portion of the left pulmonary artery (Figure 5). The right subclavian artery presented proximal occlusion, distally loading through fine periesophageal collaterals originating at the level of the descending thoracic aorta and probably also through subclavian steal. At the level of the abdominal segment of the aorta, the presence of a prominent aberrant arterial branch was highlighted, originating cranially from the emergence of the right renal artery and caudal from the emergence of the superior mesenteric artery; it presents an ascending perihepatic trajectory, then transdiaphragmatic, at the level of the costodiaphragmatic pulmonary sinus, continuing with many serpinginous arterial trajectories, which presented a progressively smaller caliber; at the same level of projection corresponding to the right costdiaphragmatic pulmonary sinus, the presence of more prominent branches of the right inferior pulmonary artery was also highlighted. In conclusion, the appearance of CTA was highly suggestive for MAPCAs between an aberrant arterial trajectory originating from the abdominal aorta and branches of the right lower pulmonary artery.

## 3. Discussion

TOF is the most common cyanotic congenital heart condition. It has been classically characterized by the combination of VSD, right ventricular outflow tract obstruction, overriding aorta, and right ventricular hypertrophy, with severe cyanosis in childhood and high morbidity and mortality [14]. However, there is a wide range of abnormalities, ranging from mild obstruction of the RV outflow tract to severe obstruction with pulmonary valve atresia. When the pulmonary arteries are very hypoplastic (as in our case) or absent, the pulmonary flow is maintained by the major aorto-pulmonary collateral arteries (MAPCAs). However, the CMR examination was not sufficient to optimally assess the complex vascular anatomy and, in particular, the collateral vessels for pulmonary supply. MRI imaging has great potential in the study of the thoracic aorta. However, compared with CT, the acquisition times are long, the spatial resolution is lower and the susceptibility of the motion artifacts is higher [15]. Particularly in our patient, we found atypical MAPCAs represented by a prominent aberrant arterial pathway coming from the abdominal aorta and the branches of the right lower pulmonary artery. All anatomical variants illustrated and described by CMR and CTA explain the clinical presentation as an adult “pink” TOF. Thus, in order to assist the multidisciplinary team in the treatment decision-making process, radiologists must have a good knowledge of TOF and its particular variants, especially since this is one of the heart diseases for which complementary imaging is most often required.

### Follow-Up

We decided to refer the patient to a surgeon for step-by-step surgical procedures, in order to prevent sudden cardiac death. Although he is still in good clinical condition, without heart failure and with normal RV function, he has hypoxemia and polycythemia. However, we considered our case very challenging for the surgical team: a large VSD, small RVOT and PA, also associated with important dilation of left pulmonary artery, and above all, with multiple atypical MAPCAs. Importantly, when the MAPCAs are significant, patients tolerate symptoms well. Due to MAPCAs, these patients develop signs of secondary pulmonary hypertension over time (due to additional systemic arterial flow in the pulmonary vascular bed) or signs of heart failure. Under these conditions, the surgery becomes much more laborious, being performed in several stages; one of these steps is the isolation of MAPCAs from their systemic origin and their central connection to the pulmonary arteries. The patient deferred cardiac surgery, because of the risks. Shared decision making is not widely practiced in our country, and it depends on many factors such as the level of education [16].

Thus, in these cases the exact knowledge of the anatomy of the vascular tree becomes essential preoperatively.

## 4. Conclusions

Our case is a very unusual combination of an adult uncorrected TOF with a large ventricular septal defect but non-obstructive stenosis at the RVOT level, small PT and both left and right PA, mild RV hypertrophy and preserved RV function, right aortic arch and distinctive aortopulmonary collaterals (“classic” thoracic aortopulmonary collateral vessels but also abdominal aortopulmonary collateral vessels). These findings explain the clinical silence and long-term survival without major events. This seems to be, to the best of our knowledge, the first case of an adult noncorrected TOF with development of collaterals from the abdominal aorta. Pulmonary CTA, as well as CMR, each provided vital details allowing a gradual approach to describing the patient’s unique anatomy. Both imaging techniques were essential in understanding the clinical silence of the TOF and discovering the adaptive collateral vessels, in order to be able to plan for surgery.

## Figures and Tables

**Figure 1 medicina-58-01011-f001:**
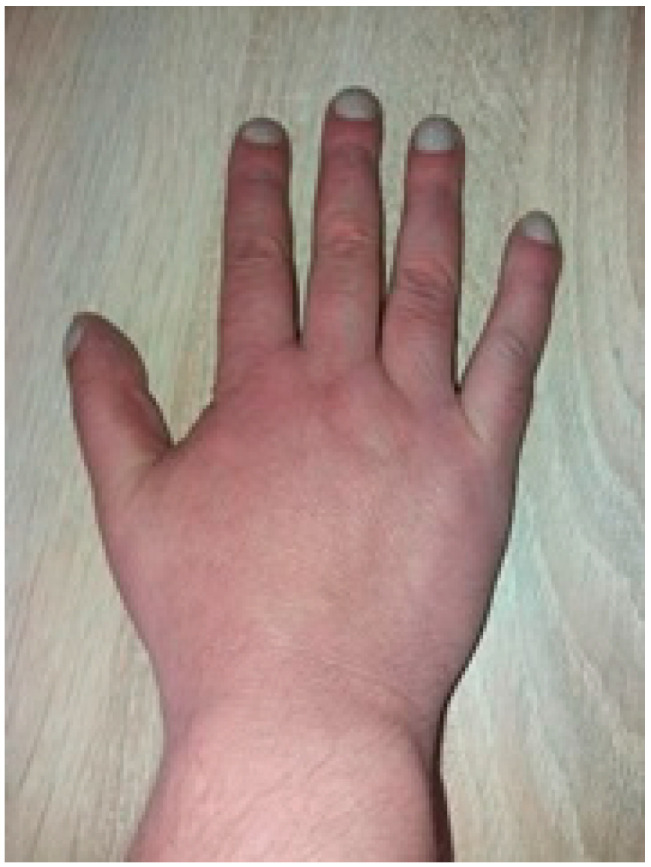
Clubbing of the fingers.

**Figure 2 medicina-58-01011-f002:**
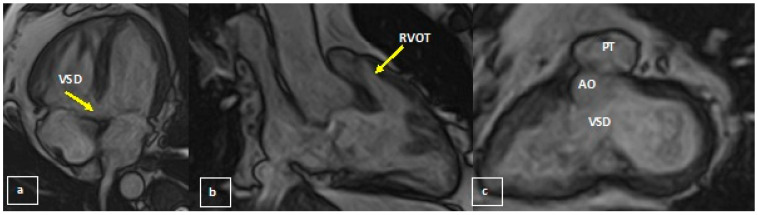
CMR: (**a**) horizontal long axis: subaortic ventricular septal defect; (**b**) RVOT: infundibular stenosis (with an RV outflow tract diameter of 13 mm); (**c**) short axis: subaortic ventricular septal defect and ascending aorta overriding the interventricular septum.

**Figure 3 medicina-58-01011-f003:**
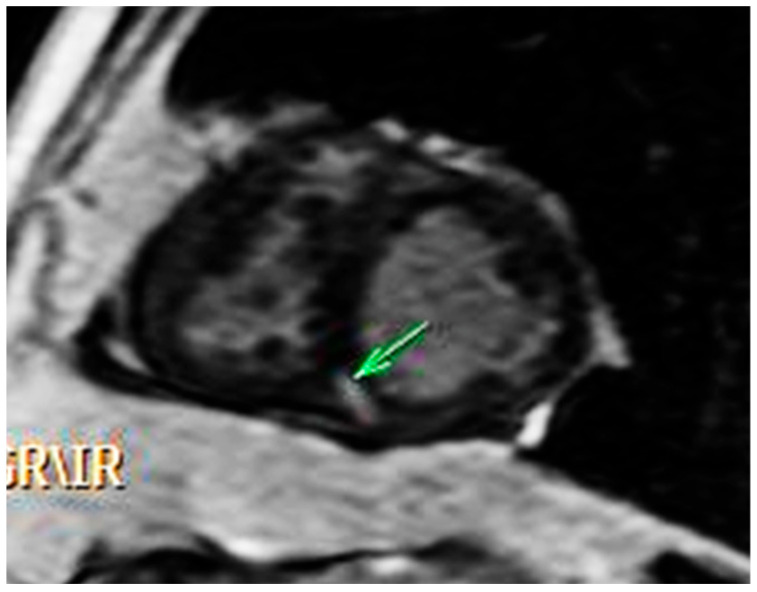
LGE: nodular contrast enhancement at the level of the posterior septal insertion of the RV free wall.

**Figure 4 medicina-58-01011-f004:**
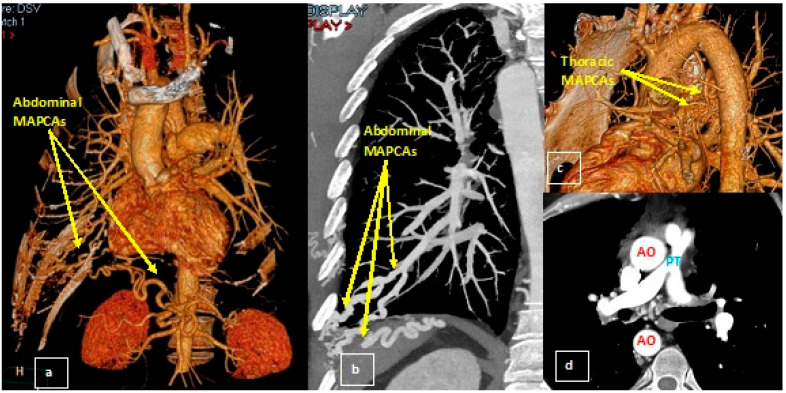
Pulmonary CTA: (**a**) 3D reconstruction and (**b**) MIP reconstruction: abdominal aortopulmonary collateral vessels; (**c**) 3D reconstruction: thoracic aortopulmonary collateral vessels; (**d**) axial section: the main pulmonary artery (PT) had a small caliber (10 mm, with an ascending aortic/pulmonary trunk ratio of 3.6), associating a small diameter of the origin of both pulmonary arteries (10 mm on the right and 8 mm on the left).

**Figure 5 medicina-58-01011-f005:**
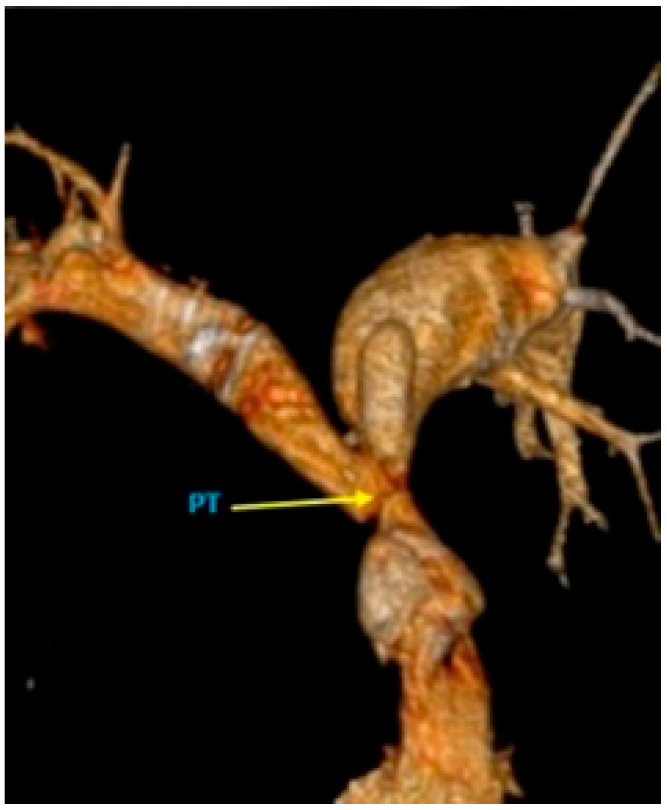
Pulmonary CTA, 3D reconstructions.

## Data Availability

Not applicable.

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
