# Peer review of "The Case of a 44-Year-Old Survivor of Unrepaired Tetralogy of Fallot, Right Aortic Arch and Abdominal Aortopulmonary Collateral Vessels"

_medicina, 2022, doi:10.3390/medicina58081011_

Round 1

Reviewer 1 Report

I have read with interest the case report by Nicula et al. They have presented a case of a 44-year-old man with unrepaired tetralogy of Fallot. The patient was mildly symptomatic.

 The case is well described and nicely illustrated.

 I have only a few minor remarks:

1.     The authors  have written: “… was sent by his employer for an annual cardiac examination.” What were the findings of previous cardiac examinations? Was ToF missed in the previous assessments?

2.     I would rather use the abbreviation “TTE” for transthoracic echocardiography instead of “ETT”

3.     Section 2.3 Diagnosis and management: I would rather write “15 mm wide” instead of “15 mm high”.

Author Response

Dear reviewer 1,

Thank you for your answers. I am happy to see that you appreciate the article.

We made the following changes:

Reviewer 1:

  1. The authors have written: “… was sent by his employer for an annual cardiac examination.” What were the findings of previous cardiac examinations? Was ToF missed in the previous assessments?

No suspicion was raised at previous routine clinical examinations. However, the diagnosis of TOF was not made during occupational medicine examinations.                                                                                           

  1. I would rather use the abbreviation “TTE” for transthoracic echocardiography instead of “ETT”.

We changed.

  1. Section 2.3 Diagnosis and management: I would rather write “15 mm wide” instead of “15 mm high”.

We changed

NB: The authors decided to change the corresponding author. From now, the corresponding author is Roxana Ciltea.

The comments and suggestions are helpful for revising and improving our manuscript. We have made all revisions. 

Best regards,

Dr. Mircea Bajdechi!

Reviewer 2 Report

It is an interesting case report of an uncorrected ToF with MAPCAs from the abdominal aorta showing the diagnostic value of CMR and CT. 

I have some comments, which in my opinion would improve the manuscript:

1. Why was CT done? Usually it is sufficient to use CMR with aortic angiography to detect MAPCAs. It is cheaper (one method used), faster and there is no radiation. Please include in the discussion. 

2. There are no numeric data from the studies provided - heart chamber volumes, ejection fraction, LV and RV mass with normal values provided. A table with this data would improve the presentation of the results. 

3. Line 48 - please use "left ventricular hypertrophy" instead of "hypertrophic cardiomyopathy" before the brackets (HT does not lead to HCM).

4. Line 97 - please change "caliber" to "diameter"

5. Lines 112-115 and 140-142 seems to be a repetition, please remove one. 

Author Response

Dear reviewer 2,

Thank you for your answers. I am happy to see that you appreciate the article.

We made the following changes:

Reviewer 2:

  1. Why was CT done? Usually it is sufficient to use CMR with aortic angiography to detect MAPCAs. It is cheaper (one method used), faster and there is no radiation. Please include in the discussion. 

MRI imaging has great potential in the study of the thoracic aorta. However, compared to CT, the acquisition times are long, the spatial resolution is lower and the susceptibility of the motion artifacts is higher (https://www.ncbi.nlm.nih.gov/pmc/articles/PMC5329815/). Therefore, in addition to its completion, a CTA examination was performed.      We included this in the discussion.

  1. There are no numeric data from the studies provided - heart chamber volumes, ejection fraction, LV and RV mass with normal values provided. A table with this data would improve the presentation of the results. 

We took as a reference the values communicated by EACVI (European Association of Cardiovascular Imaging expert consensus paper: a comprehensive review of cardiovascular magnetic resonance normal values of cardiac chamber size and aortic root in adults and recommendations for grading severity, Steffen E Petersen, Mohammed Y Khanji, Sven Plein, Patrizio Lancellotti, European Heart Journal - Cardiovascular Imaging, Volume 20, Issue 12, December 2019, Pages 1321–1331, https://doi.org/10.1093/ehjci/jez232, Published:23 September 2019)

  1. Line 48 - please use "left ventricular hypertrophy" instead of "hypertrophic cardiomyopathy" before the brackets (HT does not lead to HCM).

We changed.

  1. Line 97 - please change "caliber" to "diameter"

We changed.

  1. Lines 112-115 and 140-142 seems to be a repetition, please remove one. 

We removed.

NB: The authors decided to change the corresponding author. From now, the corresponding author is Roxana Ciltea.

The comments and suggestions are helpful for revising and improving our manuscript. We have made all revisions. 

Best regards,

Dr. Mircea Bajdechi!